# Baleen: Robust Multi-Hop Reasoning at Scale via Condensed Retrieval

**Omar Khattab**
Stanford University
okhattab@stanford.edu

**Christopher Potts**
Stanford University
cgpotts@stanford.edu

**Matei Zaharia**
Stanford University
matei@cs.stanford.edu

## Abstract

Multi-hop reasoning (i.e., reasoning across two or more documents) is a key ingredient for NLP models that leverage large corpora to exhibit broad knowledge. To retrieve evidence passages, multi-hop models must contend with a fast-growing search space across the hops, represent complex queries that combine multiple information needs, and resolve ambiguity about the best order in which to hop between training passages. We tackle these problems via Baleen, a system that improves the accuracy of multi-hop retrieval while learning robustly from weak training signals in the many-hop setting. To tame the search space, we propose *condensed retrieval*, a pipeline that summarizes the retrieved passages after each hop into a single compact context. To model complex queries, we introduce a *focused late interaction* retriever that allows different parts of the same query representation to match disparate relevant passages. Lastly, to infer the hopping dependencies among unordered training passages, we devise *latent hop ordering*, a weak-supervision strategy in which the trained retriever itself selects the sequence of hops. We evaluate Baleen on retrieval for two-hop question answering and many-hop claim verification, establishing state-of-the-art performance.

## 1 Introduction

In *open-domain* reasoning, a model is tasked with retrieving evidence from a large corpus to answer questions, verify claims, or otherwise exhibit broad knowledge. In practice, such open-domain tasks often further require *multi-hop* reasoning, where the evidence must be extracted from two or more documents. To do this effectively, a model must learn to use *partial* evidence it retrieves to bridge its way to additional documents leading to an answer. In this vein, HotPotQA [31] contains complex questions answerable by retrieving two passages from Wikipedia, while HoVer [13] contains claims that can only be verified (or disproven) by combining facts from up to four Wikipedia passages.

Table 1 illustrates this using the claim *"The MVP of [a] game Red Flaherty umpired was elected to the Baseball Hall of Fame"* from the HoVer validation set. To verify this claim, a HoVer model must identify facts spread across three Wikipedia pages: *Red Flaherty* umpired in the World Series in 1955, 1958, 1965, and 1970. The MVP of the *1965 World Series* was *Sandy Koufax*. Koufax was later elected to the Baseball Hall of Fame.

This three-hop claim illustrates three major challenges in multi-hop retrieval. First, *multi-hop queries encompass multiple information needs*; the claim above referenced facts from three disparate passages. Second, *retrieval errors in each hop propagate to subsequent hops*. This can happen if the model directly retrieves information about the Baseball Hall of Fame, confuses Red Flaherty with, say, Robert Flaherty, or singles out the MVP of, say, the *1958* World Series, which Flaherty also umpired. Third, due to the dependency between hops, *retrievers must learn an effective sequence of hops*, where previously-retrieved clues lead to other relevant passages. These inter-passage dependencies

35th Conference on Neural Information Processing Systems (NeurIPS 2021).

Table 1: An example (multi-hop) claim $Q_0$ to verify and an illustration of *condensed retrieval* queries $Q_t$ after each hop $t \geq 1$. Page titles are in bold. We shorten the Wikipedia sentences for presentation.

| | |
|---|---|
| $Q_0$ | The MVP of [a] game Red Flaherty umpired was elected to the Baseball Hall of Fame. |
| $Q_1$ | The MVP of [a] game Red Flaherty umpired was elected to the Baseball Hall of Fame. **Red Flaherty**: He umpired in World Series 1955, 1958, 1965, and 1970. |
| $Q_2$ | The MVP of [a] game Red Flaherty umpired was elected to the Baseball Hall of Fame. Red Flaherty: He umpired in World Series 1955, 1958, 1965, and 1970. **1965 World Series**: It is remembered for MVP Sandy Koufax. |
| $Q_3$ | The MVP of [a] game Red Flaherty umpired was elected to the Baseball Hall of Fame. Red Flaherty: He umpired in World Series 1955, 1958, 1965, and 1970. 1965 World Series: It is remembered for MVP Sandy Koufax. **Sandy Koufax:** He was elected to the Baseball Hall of Fame. |

can be non-obvious for *many-hop* problems with three or more passages, and are often left unlabeled, as it can be expensive to annotate one (or every) sequence in which facts could be retrieved.

These challenges call for highly expressive query representations, robustness to retrieval errors, and scalability to many hops over massive document collections. Existing systems fall short on one or more of these criteria. For instance, many state-of-the-art systems rely on bag-of-words [23] or single-vector dot-product [29] retrievers, whose capacity to model an open-domain question is limited [17], let alone complex multi-hop queries. Furthermore, existing systems embed trade-offs when it comes to "hopping": they employ brittle greedy search, which limits recall per hop; or they employ beam search over an exponential space, which reduces scalability to many hops; or they assume explicit links that connect every passage with related entities, which ties them to link-structured corpora. Lastly, to order the hops, many systems use fragile supervision heuristics (e.g., finding passages whose page titles appear in other passages) tailored for particular datasets like HotPotQA.

We tackle these problems with **Baleen**,[1] a scalable multi-hop reasoning system that improves accuracy and robustness. We introduce a *condensed retrieval* architecture, where the retrieved facts from each hop are summarized into a short context that becomes a part of the query for subsequent hops, if any (Table 1). Unlike beam search, condensed retrieval allows effective scaling to many hops, and we find that it complements greedy search (i.e., taking the best passage per hop) in improving recall considerably. We then tackle the complexity of queries by proposing a *focused late interaction* passage retriever (**FLIPR**), a robust learned search model that allow different parts of the same query representation to match disparate relevant passages. FLIPR inherits the scalability of the vanilla late interaction paradigm of ColBERT [16] but uniquely allows the same query to exhibit tailored matching patterns against each target passage. Lastly, we devise *latent hop ordering*, a weak-supervision strategy that uses the retriever itself to select effective hop paths.

We first test Baleen on the two-hop HotPotQA benchmark, finding evidence of saturation in retrieval: we achieve 96.3% answer recall in the top-20 retrieved passages, up from 89.4% for existing work. We then test Baleen's ability to scale accurately to more hops, reporting our main results using the recent many-hop HoVer task. We build a strong *many-hop* baseline model that combines recent results from the open-domain question answering [17] and multi-hop [23] literatures. This baseline combines ColBERT retrieval and standard greedy search with an ELECTRA [5] re-ranker, and generalizes typical text-based heuristics to order the hops for training. After verifying its strong results on HotPotQA, we show that it outperforms the official TF-IDF + BERT baseline of HoVer by over 30 points in retrieval accuracy. Against this strong baseline itself, Baleen improves passage retrieval accuracy by another 17 points, raising it to over 90% at $k = 100$ passages. Baleen also improves the evidence extraction F1 score dramatically, outperforming even the *oracle retrieval* + BERT results by Jiang et al. [13]. Our ablations (§5.4) show that Baleen's FLIPR retriever, condenser architecture, and latent supervision are essential to its strong performance.

---

[1]`https://github.com/stanford-futuredata/Baleen`
In marine biology, "baleen" refers to the filter-feeding system that baleen whales use to capture small organisms, filtering out seawater. So too does our system seek to capture relevant facts from a sea of documents.

## 2 Background and related work

**Open-domain reasoning** There is significant recent interest in NLP models that can solve tasks by retrieving evidence from a large corpus. The most popular such task is arguably open-domain question answering (OpenQA; Chen et al. [2]), which extends the well-studied machine reading comprehension (MRC) problem over supplied question–passage pairs to answering questions given only a large text corpus like Wikipedia. Other open-domain tasks span claim verification (e.g., FEVER; Thorne et al. [24]), question generation (e.g., with Jeopardy as in Lewis et al. [19]), and open dialogue (e.g., Wizard of Wikipedia; Dinan et al. [8]), among others. Many of these datasets are compiled in the recently-introduced KILT benchmark [21] for *knowledge-intensive* tasks. Among models for these tasks, the most relevant to our work are OpenQA models that include *learned* retrieval components. These include ORQA [18], REALM [10], and DPR [15]. Lewis et al. [19] introduced RAG, which by virtue of a seq2seq architecture can tackle OpenQA as well as other open-domain problems, including claim verification and question generation.

**Multi-hop open-domain reasoning** Most of the open-domain tasks from §2 can be solved by finding *one* relevant passage in the corpus, often by design. In contrast, a number of recent works explore multi-hop reasoning over multiple passages. These include QAngaroo [27] and 2WikiMulti-HopQA [11], among others. While thus "multi-hop", these tasks supply the relevant passages for each example (possibly with small set of "distractor" candidates), and thus do not require retrieval from a large corpus. To our knowledge, HotPotQA was the first large-scale open-domain multi-hop task, particularly in its retrieval-oriented "fullwiki" setting. HotPotQA catalyzed much follow-up research. However, it is limited to only two-hop questions, many of which may not require strong multi-hop reasoning capabilities (see Chen and Durrett [3]; Wang et al. [26]), softening the retrieval challenges discussed in §1. Very recently, Jiang et al. [13] introduced the HoVer many-hop verification dataset, which contains two-, three-, and four-hop examples. HoVer contains just over 18,000 training examples, about $5\times$ smaller than HotPotQA, adding to the challenge posed by HoVer.

**Multi-hop open-domain models** To conduct the "hops", many prior multi-hop systems (e.g., Asai et al. [1]; Zhao et al. [32]) assume explicit links that connect every passage with related entities. We argue that this risks tying systems to link-structured knowledge bases (like Wikipedia) or producing brittle architectures tailored for datasets constructed by following hyperlinks (like HotPotQA). Recently, Xiong et al. [29] and Qi et al. [23] introduce MDR and IRRR, state-of-the-art systems that assume no explicit link structure. Instead, they use an *iterative retrieval* paradigm—akin to that introduced by Das et al. [6] and Feldman and El-Yaniv [9]—that retrieves passages relevant to the question, reads these passages, and then formulates a new query for another hop if necessary. We adopt this iterative formulation and tackle three major challenges for multi-hop retrieval.

**ColBERT: late interaction paradigm** Most learned-retrieval systems for OpenQA (§2) encode every query and passage into a *single* dense vector. Khattab and Zaharia [16] argue that such single-vector representations are not sufficiently expressive for retrieval in many scenarios and introduce *late interaction*, a paradigm that represents every query and every document at a finer granularity: it uses a vector *for each constituent token*. Within this paradigm, they propose ColBERT, a state-of-the-art retriever wherein a BERT encoder embeds the query into a matrix of $N$ vectors (given $N$ tokens) and encodes every passage in the corpus as matrix of $M$ vectors (for $M$ tokens per passage). The passage representations are query-independent and thus are computed offline and indexed for fast retrieval.

During retrieval with query $q$, ColBERT assigns a score to a passage $d$ by finding the maximum-similarity (MaxSim) score between each vector in $q$'s representation and *all* the vectors of $d$ and then summing these MaxSim scores. This MaxSim-decomposed interaction enables scaling to massive collections with millions of passages, as the token-level passage vectors can all be indexed for fast nearest-neighbor search. Khattab and Zaharia [16] find this fine-grained matching to outperform single-vector representations, a finding extended to open-domain question answering by Khattab et al. [17]. In this work, we propose FLIPR (§3.1), a retriever with an improved late interaction mechanism for complex multi-hop queries. Whereas existing retrievers, whether traditional (e.g., TF-IDF) or neural [18; 15; 16], seek passages that match *all* of the query, multi-hop queries can be long and noisy and need to match disparate contexts. FLIPR handles this explicitly with *focused* late interaction.

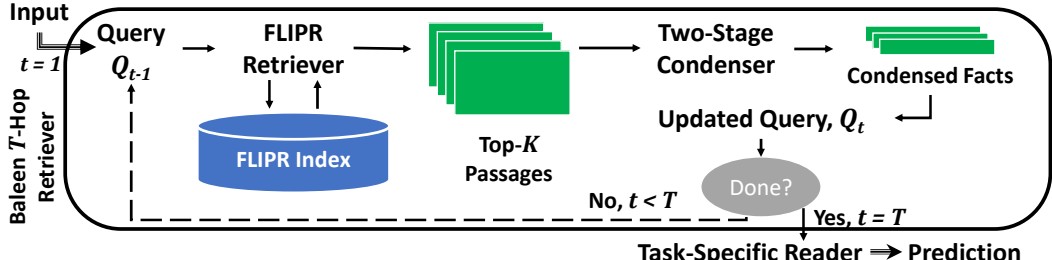

Figure 1: The architecture of Baleen with iterative retrieval and condensing. The model begins with iteration $t = 1$, provided initial user input $Q_0$. For every $t$, FLIPR takes in $Q_{t-1}$ and retrieves the top-$K$ passages, the condenser summarizes the relevant facts, and an updated query $Q_t$ is produced. When $t = T$, $Q_T$ is fed to the reader, which outputs a final prediction.

**Relevance-guided supervision** Khattab et al. [17] propose an iterative strategy for weakly-supervising retrievers called relevance-guided supervision (RGS). RGS assumes that no labeled passages are supplied for training the retriever. Instead, every training question is associated with a short answer string whose presence in a passage is taken as a weak signal for relevance. RGS starts with an off-the-shelf retriever and uses it to collect the top-$k$ passages for each training question, dividing these passages into positive and negative examples based on inclusion of the short answer string. These examples are used to train a stronger retriever, which is then used to repeat this process one or two times. We draw inspiration from RGS when designing latent hop ordering: unlike RGS, we *do* have gold-labeled passages for training but we crucially have multiple hops, whose order is not given. Lastly, our supervision for the *first hop* shares inspiration with GoldEn [22]. GoldEn uses a bag-of-words model—and thus effectively term overlap—to identify the "more easily retrieved" passage among just two for the two-hop task HotPotQA.

## 3 Baleen

We now introduce Baleen, which uses an iterative retrieval paradigm to find relevant facts in $T \geq 1$ successive hops. On HotPotQA (§4) and HoVer (§5), we use Baleen with $T = 2$ and $T = 4$, respectively. As illustrated in Table 1, the input to Baleen is a textual *query* $Q_0$ like a question to answer or a claim to verify. The goal of hop $t$ is to take in query $Q_{t-1}$ and output an updated query $Q_t$ containing the initial input query *and* pertinent facts extracted from the $t$ hops.

The Baleen architecture is depicted in Figure 1. In every hop, FLIPR (§3.1) uses $Q_{t-1}$ to retrieve $K$ passages from the full corpus. These passages are fed into a two-stage *condenser* (§3.3), which reads these passages and extracts the most relevant *facts*, which we model as individual sentences. The facts are collected into a single sequence and added to $Q_t$ for the subsequent hop, if any. Once all hops are complete, Baleen's task-specific *reader* processes $Q_T$, which now contains the query and all condensed facts, to solve the downstream task. If desired, the top-$k$ passages from each hop can also be collected and fed to a downstream model after retrieval is complete. Baleen's retriever, condenser, and reader are implemented as Transformer encoders [25] and are trained individually. We use BERT-base [7] for FLIPR and, like Qi et al. [23] and Xiong et al. [29], use ELECTRA-large [5] for the other components. For supervision, we introduce the *latent hop ordering* scheme (§3.2).

### 3.1 FLIPR: focused late interaction

The FLIPR encoders and interaction mechanism are shown in Figure 2. Our query encoder reads $Q_{t-1}$ and outputs a vector representation of every token in the input. Each query embedding interacts with all passage embeddings via a MaxSim operator, permitting us to inherit the efficiency and scalability of ColBERT [16]. While vanilla ColBERT sums all the MaxSim outputs indiscriminately, FLIPR considers only the strongest-matching query embeddings for evaluating each passage: it sums only the top-$\hat{N}$ partial scores from $N$ scores, with $\hat{N} < N$.

We refer to this top-$k$ filter as a "focused" interaction. Intuitively, it allows the same query to match multiple relevant passages that are contextually unrelated, by aligning—during training and inference—a different subset of the query embeddings with different relevant passages. These

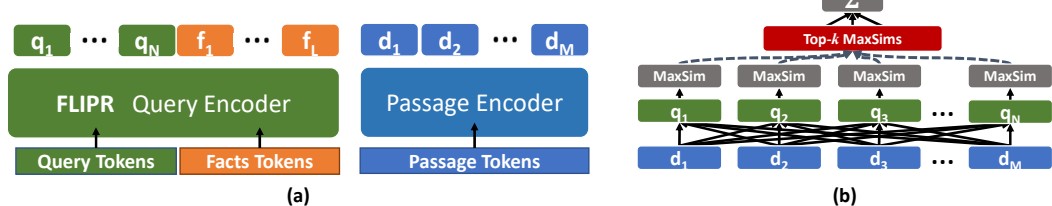

Figure 2: The proposed FLIPR retriever. Sub-figure (a) depicts our encoders, which represent every query and every passage as a *set* of embeddings. Given query $q$ and document $d$, sub-figure (b) illustrates our *focused* late-interaction mechanism. It finds the maximum-similarity score of each query embedding $\mathbf{q}_i$ and then *focuses* on the top-$k$ matches only. This is analogously applied to the fact embeddings $\mathbf{f}_j$ (in orange), and both scores are finally added.

important subsets of the query are not always clear in advance. For instance, $Q_1$ in Table 1 should find the page about the 1965 World Series. This requires ignoring many distractions (e.g., World Series 1970 and Baseball Hall of Fame) that seem important in isolation while focusing on other phrases, namely a prominent "MVP" and the World Series 1965, a distinction evident only *after* finding the document and attempting to match it against the query.

FLIPR applies this mechanism over the representation of the query and the facts separately, keeping the top-$\hat{N}$ and top-$\hat{L}$ from $N$ and $L$ embeddings, respectively. More formally, the FLIPR score $S_{q,d}$ of a passage $d$ given a query $q$ is the summation $S_{q,d} = S_{q,d}^Q + S_{q,d}^F$ over the scores corresponding to the original query and to the facts. Define $M_i^Q = \max_{j=1}^m E_{q_i} \cdot E_{d_j}^T$, that is, the maximum similarity of the $i^{th}$ query embedding against the passage embeddings. The partial score $S_{q,d}^Q$ is computed as

$$S_{q,d}^Q = \sum_{i=1}^{\hat{N}} \text{top}_{\hat{N}} \left\{ M_i^Q : i \in [N] \right\} \tag{1}$$

where $\text{top}_{\hat{N}}$ is an operator that keeps the largest $\hat{N}$ values in a set. We define $S_{q,d}^F$ similarly, using the fact matches $M_i^F$ instead of $M_i^Q$.

We confirm the gains of FLIPR against the state-of-the-art ColBERT retriever in §5.4. By decomposing retrieval into token-level maximum similarities and pre-computing document representations, FLIPR maintains the scalability of the late interaction paradigm of ColBERT, reducing the candidate generation stage of retrieval into highly-efficient approximate $k$-nearest neighbor searches. See Appendix §B.1 for the implementation details of FLIPR.

### 3.2 Supervision: latent hop ordering

For every training example, our datasets supply *unordered* gold passages. However, dependencies often exist in retrieval as Table 1 illustrates: a system needs to retrieve and read the page of *Red Flaherty* to deduce that it needs information about *1965 World Series*. Such dependencies can be complex for 3- and 4-hop examples, as in HoVer, especially as multiple passages may be appropriate to retrieve together in the same hop.

We propose a generic mechanism for identifying the best gold passage(s) to learn to retrieve next. The key insight is that among the gold passages, a weakly-supervised retriever with strong inductive biases would reliably "prefer" those passages it can more naturally retrieve, given the hops so far. Thus, given $Q_{t-1}$, we train the retriever with *every* remaining gold passage and, among those, label as positives (for $Q_{t-1}$) only those ranked highly by the model. We subsequently use these weakly-labeled positives to train another retriever for all the hops.

Latent hop ordering is summarized in Algorithm 1, which assumes we have already trained a single-hop (i.e., first-hop) retriever $R_1$ in the manner of relevance-guided supervision (see §2). To start, we use $R_1$ to retrieve the top-$k$ (e.g., $k = 1000$) passages for each training question (Line 2). We then divide these passages into positives $P_1$ and negatives $N_1$ (Line 3). Positives are the highly-ranked gold passages (i.e., the intersection of gold passages and the top-$\hat{k}$ passages with, e.g., $\hat{k} = 10$) and

**Algorithm 1:** Latent Hop Ordering, a simplified procedure

---

**Input:** Training queries & unordered gold passages; Corpus of all passages; A single-hop retriever, $R_1$
**Output:** A multi-hop retriever, $\hat{R}$

1  $Q_0 \leftarrow$ the original training queries;
2  $Ranking_1 \leftarrow R_1.retrieve(Q_0)$;
3  $P_1, N_1 \leftarrow ExtractPositives(Ranking_1)$;
4  $Q_1 \leftarrow Q_0.expand(OracleFacts(P_1))$;
5  **for** hop $t \leftarrow 2$ **to** $T$ **do**
6  $\quad$ Train a new retriever $R_t$ using queries $Q_{t-1}$ and negatives $N_{t-1}$. For positives, use gold passages not part of any $P_s$ for $s < t$.
7  $\quad$ $Ranking_t \leftarrow R_t.retrieve(Q_{t-1})$;
8  $\quad$ $P_t, N_t \leftarrow ExtractPositives(Ranking_t)$; $Q_t \leftarrow Q_{t-1}.expand(OracleFacts(P_t))$;
9  **end**
10  For every hop $t \in [T]$, designate $Q_{t-1}$ as queries, $P_t$ as positives, and $P_t$ as negatives.
11  Train a final retriever $\hat{R}$ using this combined training data for all hops.
12  **return** $\hat{R}$

---

negatives are the non-gold passages. Subsequently, we expand first-hop queries with the oracle facts from $P_1$ to obtain the queries for the second hop (Line 4).

We train a second-hop retriever $R_2$ using the aforementioned queries and negatives (Line 6). As we do not have second-hop positives, as a form of weak supervision, we train the retriever with *all* gold passages (per query) besides those already in $P_1$. Once trained, we use $R_2$ to discover the second-hop positives $P_2$, to collect negatives $N_2$, and to expand the queries (Lines 7 and 8). We repeat this procedure for the third hop onward. Once this iterative procedure is complete, we have bootstrapped positives (and negatives) corresponding to every retrieval hop for each query. We combine these sets to train a single multi-hop retriever (Lines 10 and 11), which takes a query $Q_{t-1}$ and learns to discriminate the positives in $P_t$ from the negatives in $N_t$ for every $t \in [T]$. In §5.4, we indeed find that this *generic* latent procedure outperforms a typical hand-crafted heuristic that relies on trying to match titles against the text.

### 3.3 Condenser: per-hop fact extraction

After each hop, the condenser proceeds in two stages. In the first stage, it reads $Q_{t-1}$ and each of the top-$K$ passages retrieved by FLIPR. The input passages are divided into their constituent sentences and every sentence is prepended with a special token. The output embedding corresponding to these sentence markers are scored via a linear layer, assigning a score to each sentence in every retrieved passage. We train the first-stage condenser with a cross-entropy loss over the sentences of two passages, a positive and a negative.

Across the $K$ passages, the top-$k$ facts are identified and concatenated into the second-stage condenser model. As above, we include special tokens between the sentences, and the second-stage condenser assigns a score to every fact while attending jointly over all of the top-$k$ facts, allowing a direct comparison within the encoder. Every fact whose score is positive is selected to proceed, and the other facts are discarded. We train the second-hop condenser over a set of 7–9 facts, some positive and others (sampled) negative, using a linear combination of cross-entropy loss for each positive fact (against all negatives) and binary cross-entropy loss for each individual fact.

In practice, this condensed architecture offers novel tradeoffs against a re-ranking pipeline. On one hand, condensed retrieval scales better to more hops, as it represents the facts from $K$ long passages using only a few sentences. This may provide stronger interpretability, as the inputs to the retriever (e.g., the example in Table 1) become more focused. On the other hand, re-ranking requires only passage-level labels, which can be cheaper to collect, and provides the retriever with broader context, which can help resolve ambiguous references. Empirically, we find the two approaches to be complementary. In §5.4, we show that condensing is competitive with re-ranking (despite much shorter context; Appendix §C) and that a single retriever that combines both pipelines together substantially outperforms a standard rerank-only pipeline.

Table 2: Retrieval results on HotPotQA-fullwiki (dev) showing saturation in finding the gold pair of passages and passages with the answer string. Baleen and its ablation baseline ColBERT-Hop exceed 90–93% P-R@20 and 94–96% Ans-R@20, largely reducing the problem to *reading* the retrieved passages. We obtain P-EM for MDR from Xiong et al. [29] and for IRRR [23] from the authors.

| Model | Retriever | Architect. | Supervis. | P-EM | P-R@20 | Ans-R@20 |
|---|---|---|---|---|---|---|
| MDR | DPR | Beam | Heuristic | 81.2 | 82.9 | 89.4 |
| IRRR | ELECTRA/BM25 | Rerank | Heuristic | 84.1 | - | - |
| ColBERT-Hop (ablation) | ColBERT | Rerank | Heuristic | 85.2 | 90.3 | 94.7 |
| Baleen | FLIPR | Condense | Latent | **86.7** | **93.3** | **96.3** |

## 3.4 Reader: task-specific processing

After all hops are conducted, a reader is used to extract answers or verify the claim. We use a standard Transformer encoder, in particular ELECTRA-large [5], which we feed the final output $Q_t$ of our multi-hop retrieval and condensing pipeline. We train for question answering similar to Khattab et al. [17] or claim verification similar to Jiang et al. [13].

## 4 Diagnosing retrieval saturation on HotPotQA

Before beginning our primary evaluation on the many-hop HoVer benchmark, we first investigate the capacity of HotPotQA in reflecting the challenges on multi-hop retrieval (§1). We use HotPotQA's "fullwiki" setting, where the input to the model is a question whose answer requires retrieving two passages from Wikipedia. Similar to related work [29], we report passage-level exact-match (EM) and Recall@$k$ (R@$k$) on the development set. These are the percentage of questions for which the model correctly identifies the two gold passages and for which the model's top-$k$ retrieved passages contain both gold passages, respectively.[2] We also report Answer-Recall@$k$, the percentage of questions whose short answer string is found in the top-$k$ passages (excluding yes/no questions). Here, the top-20 passages from Baleen are simply the union of the top-10 passages retrieved during the first hop and the top-10 passages from the second hop, without duplicates.

We compare against the state-of-the-art models MDR and IRRR. We additionally compare with *ColBERT-Hop*, a strong baseline that ablates Baleen's retriever, condenser, and supervision: ColBERT-Hop uses the state-of-the-art neural retriever ColBERT (see §2), employs a typical re-ranker architecture, and uses heuristic supervision. The results are shown in Table 2. Baleen demonstrates strong results, finding both gold passages in over 93% of all questions and finding passages that contain the short answer string for over 96% of all questions in just the top $k$=20 passages. We argue that this effectively reduces the open-retrieval task to a narrow distractor-setting one, in which *reader* models are supplied with only a few passages (ones that are (nearly) guaranteed to contain the answer) and are then expected to extract the answer from just those passages.

In other words, such high recall largely shifts the competition in downstream quality to the reader, since it almost always has access to the answer in the top-$k$ set and is thus responsible for the majority of downstream failures. While reading the retrieved passages may still pose additional, important challenges beyond retrieval that remain open, it should be just one of the many challenging aspects of multi-hop reasoning at scale. This motivates our focus on the many-hop HoVer task.

## 5 Main evaluation on HoVer

We now conduct our primary evaluation, applying Baleen on the many-hop claim verification dataset HoVer, following the evaluation protocol of Jiang et al. [13]. First, we study the retrieval performance of Baleen in §5.2. Subsequently (§5.3), we investigate its sentence extraction and claim verification performance, which constitutes a test of overall end-to-end system effectiveness. We compare a

---

[2]For MDR, we use the authors released retrieval to compute P-R@20. The authors originally report 80.2, but we do not penalize the MDR retriever for finding both gold passages split across different pairs.

Table 3: Our main passage retrieval results on HoVer. To compare with Jiang et al. [13], the Retrieval@100 results are for *supported* claims. The other results are for all claims. *Marked result rows are from Jiang et al. The table reports fine-grained dev-set scores, and we submitted to the organizers and obtained **64.6** Psg-EM and **88.9** Psg-F1 for Baleen on the held-out test set.

| Model / # Hops | Retrieval@100 | | | | Passage EM | | | | Passage F1 | | | |
| --- | --- | --- | --- | --- | --- | --- | --- | --- | --- | --- | --- | --- |
| | **All** | **2** | **3** | **4** | **All** | **2** | **3** | **4** | **All** | **2** | **3** | **4** |
| TF-IDF + BERT* | 44.6 | 80.0 | 39.2 | 15.6 | 12.5 | 34.0 | 5.8 | 1.0 | 60.2 | 69.9 | 58.2 | 53.4 |
| ColBERT-Hop | 74.8 | 95.8 | 77.9 | 47.6 | - | - | - | - | - | - | - | - |
| Baleen | **92.2** | **97.7** | **93.1** | **85.1** | **63.6** | **75.8** | **62.5** | **52.6** | **89.2** | **90.2** | **89.9** | **86.8** |
| Oracle + BERT* | - | - | - | - | 34.0 | 50.9 | 28.1 | 26.2 | 80.6 | 81.7 | 79.1 | 82.2 |
| Human* | - | - | - | - | 77.0 | 85.0 | 82.4 | 65.8 | 93.5 | 92.5 | 95.3 | 91.4 |

four-hop Baleen system against the official baseline and the strong ColBERT-Hop baseline from §4.[3] See Appendix §B for implementation details and hyperparameters.

## 5.1 Task description

The input to the model is a claim, which is often one or a few sentences long. The model is to attempt to verify this claim, outputting "Supported" if the corpus contains enough evidence to verify the claim, or otherwise "Unsupported" if the corpus contradicts—or otherwise fails to support—this claim. Alongside this binary label, the model must extract facts (i.e., sentences) that justify its prediction. In particular, HoVer contains two-, three-, and four-hop claims that require facts from 2–4 different Wikipedia pages, and the model must find and extract one or more supporting facts from *each* of these pages for every claim. Models are not given the number of hops per claim.

## 5.2 Evaluating retrieval effectiveness

Table 3 reports the retrieval and passage extraction quality of the systems. We report three metrics, each sub-divided by the number of hops of each claim. The first is *Retrieval@100*, which is the percentage of claims for which the system retrieves *all* of the relevant passages within the top-100 results. To compare with Jiang et al. [13], we report the Retrieval@100 results only for supported claims. All other metrics are reported for all claims. The second metric is *Passage EM*, which is the percentage of claims for which the system can provide the *exact* set of relevant passages. The third metric is *Passage F1*, which uses the standard F1 measure to offer a relaxed version of EM.

At the top of the table, the TF-IDF baseline retrieves top-100 results in one round of retrieval. At the bottom of the table, we include the results reported by Jiang et al. [13] for their BERT ranking model when supplied with oracle retrieval, and also their reported human performance. We show Baleen's performance after four hops, where Baleen's FLIPR retrieves 25 passages in each hop. Note that we exclude passages retrieved in hop $t$ from the retrieval in further hops $t' > t$. As the table shows, Baleen achieves strong results, outperforming the official baseline model by 47.6 points in Retrieval@100, 51.1 points in Passage EM, and 29.0 points in Passage F1. In fact, Baleen's performance also consistently exceeds "Oracle + BERT" at passage extraction.

## 5.3 Evaluating end-to-end effectiveness

Next, we evaluate the performance of fact/sentence extraction and thereby the end-to-end claim verification of Baleen. The results are in Table 4, where *Sentence EM* and *Sentence F1* are defined similar to the corresponding metrics in Table 3 but focus on sentence-level extraction. Moreover, *Verification Accuracy* is the percentage of claims that are labelled correctly as "Supported" or "Unsupported". In Table 4, we see that Baleen's sentence-level results mirror its document-level results, with large performance gains across the board against the baseline model. Baleen also outperforms the oracle-retrieval BERT model of Jiang et al. [13], emphasizing the strong retrieval

---

[3]At the time of writing, MDR's open-source implementation only supports two-hop retrieval and the IRRR implementation is not publicly available. Moreover, both Xiong et al. [29] and Qi et al. [23] explore training only with *two* gold passages, though IRRR includes evaluation on a yet-unreleased 3-hop test set.

Table 4: Our main sentence extraction results on HoVer. *Marked results are from Jiang et al. [13]. The table reports fine-grained results on the dev set and, where available, includes "dev/test" scores separated by a slash for the held-out test set score. In our test evaluation on the HoVer leaderboard, Baleen increases the overall "HoVer Score" metric from **15.3** for the baseline to **57.5**.

| Model / # of Hops | Sentence EM | | | | Sentence F1 | | | | Verification Accuracy |
|---|---|---|---|---|---|---|---|---|---|
| | **All** | **2** | **3** | **4** | **All** | **2** | **3** | **4** | |
| TF-IDF + BERT* | 4.8/4.5 | 13.6 | 1.9 | 0.2 | 50.6/49.5 | 57.2 | 49.8 | 45.0 | 73.7 |
| Baleen 1-hop | 19.7 | 40.9 | 15.4 | 4.3 | 72.3 | 77.5 | 72.4 | 66.4 | - |
| Baleen 2-hop | 37.0 | 46.9 | 35.7 | 28.4 | 80.8 | 81.2 | 81.8 | 78.7 | - |
| Baleen 3-hop | 38.9 | 47.1 | 37.0 | 33.2 | 81.4 | **81.2** | 82.3 | **80.0** | - |
| Baleen 4-hop | **39.2**/39.8 | **47.3** | **37.7** | **33.3** | **81.5**/80.4 | **81.2** | **82.5** | **80.0** | **84.5**/84.9 |
| Oracle + BERT* | 19.9 | 25.0 | 18.4 | 17.1 | 71.9 | 68.3 | 71.5 | 76.4 | 81.2 |
| Human* | 56.0 | 75.0 | 73.5 | 42.1 | 88.7 | 86.5 | 93.1 | 87.3 | 88.0 |

Table 5: Retrieval@100 comparison between Baleen and five ablation settings.

| | Model | All | 2 | 3 | 4 |
|---|---|---|---|---|---|
| | **Architecture Ablations** | | | | |
| **A** | Baleen_{CONDENSE} (main architecture) | 92.2 | 97.7 | 93.1 | 85.1 |
| **B** | Baleen_{RERANK} | 91.2 | 98.7 | 93.0 | 80.0 |
| **C** | Baleen_{HYBRID} | **94.5** | **99.2** | **94.4** | **90.0** |
| | **Retrieval Ablations** (over arch. **A**) | | | | |
| **D** | w/o FLIPR (uses ColBERT retrieval modeling) | 87.4 | 97.3 | 89.5 | 73.4 |
| **E** | w/o LHO (uses a recursive title/passage-overlap heuristic for ordering) | 84.8 | 97.1 | 88.3 | 65.6 |
| | **Full Model Ablation** (over arch. **B**) | | | | |
| **F** | ColBERT-Hop (w/o FLIPR, LHO) | 74.8 | 95.8 | 77.9 | 47.6 |

and condensing performance of Baleen. In the leaderboard test-set evaluation (scores expected to be posted soon), Baleen increases the overall "HoVer Score" metric from 15.3 for the baseline to 57.5.

## 5.4 Ablation studies

Lastly, Table 5 reports various settings and ablations of Baleen, corresponding to its contributions. Here, models **A** and **F** are simply the four-hop Baleen and ColBERT-Hop in the previous experiments, respectively. Models **B** and **C** test the effect of different architectural decisions on Baleen. In particular, model **B** replaces our condensed retrieval architecture with a simpler re-ranking architecture (Appendix §B.2), which reflects a common design choice for multi-hop systems. In this case, FLIPR is given the top-ranked passage as context in each hop, helping expose the value of extracting facts in condensed retrieval, which exhibits much shorter contexts as we show in Appendix §C. Second, model **C** trains a FLIPR retriever that uses both a condenser pipeline and a reranking pipeline independently, and then retains the overall top-100 unique passages retrieved (Appendix §B.2). The results suggest that condensing offers complimentary gains, and the hybrid model attains the highest scores.

Moreover, models **D** and **E** test modifications to the retriever, while using the condenser of model **A**. Specifically, model **D** replaces our FLIPR retriever with a simpler ColBERT retriever, while retaining all other components of our architecture. This allows us to contrast our proposed focused late interaction with the "vanilla" late interaction of ColBERT. Further, model **E** replaces our weak-supervision strategy for inferring effective hop orders with a hand-crafted rule (Appendix §B.3) that deliberately exploits a construction bias in HotPotQA and HoVer, namely that passage titles can offer a strong signal for hop ordering.

# 6 Discussion

**Research limitations**   This work investigates multi-hop reasoning at a large scale by focusing on the challenges presented by retrieval in this setting. While *reading* the retrieved (or supplied) passages to solve a downstream task is another important aspect, one that has received more attention in the literature, we do not attempt to advance the state of the art approaches for multi-hop reader models. The majority of multi-hop retrieval work to date has focused on two-hop retrieval. We leverage the recently-released HoVer dataset to scale our investigation to four retrieval hops, and Baleen allows for an arbitrary number of hops in principle, but more evaluation is needed before we can claim generality to an arbitrary number of hops. We propose a condenser architecture that summarizes contexts extractively. This architectural decision allows us to scale easily to many hops as it condenses the hop information into a short context, but doing so relies on the availability on sentence-level training information (as in HotPotQA and HoVer) and extracting sentences might in principle lose important context. We expect future work to be able to infer the sentence-level labels in a latent manner and we are excited about work like decontextualization [4] that tries to retain important context for standalone sentences. Lastly, considering that learning neural retrievers for large-scale many-hop tasks is a recently-emerging topic, we had to design our own strong baseline to strengthen our HoVer evaluation, after confirming that our baseline was very strong on HotPotQA.

**Environmental and societal impact**   By turning to retrieval from text corpora (see §2) as a mechanism to find and use knowledge, we seek to build scalable and efficient models that can reason and solve downstream tasks with concrete, checkable provenance from text sources and without growing the models' number of trainable parameters dramatically. We use Wikipedia, which has favorable licensing (generally under CC BY-SA 3.0), and publicly-released datasets HoVer and HotPotQA (CC BY-SA 4.0 licenses). HotPotQA and HoVer are crowd-sourced, and we expect models trained on them to reflect the biases of the underlying data. Moreover, automated systems for answering questions or verifying claims can have misleading outputs or may even be abused. That said, we believe that focusing on retrieval and extractive models as a way to help users find information can offer net gains to society, especially in contrast with large generative models that may hallucinate or make statements ungrounded in a human-written corpus.

# 7 Conclusion

In this paper, we propose Baleen, a system for multi-hop reasoning that tackles the complexity of multi-hop queries with the *focused late interaction* mechanism for retrieval and mitigates the exponential search space problem by employing an aggressive *condensed retrieval* pipeline in every hop, which consolidates the pertinent information retrieved throughout the hops into a relatively short context while preserving high recall. Moreover, Baleen deals with the difficulty of ordering passages for many-hop supervision via *latent hop ordering*. By employing a strong retriever, incorporating a condenser, and avoiding brittle heuristics, Baleen can robustly learn from limited training signal. We evaluated Baleen's retrieval on HotPotQA and the recent many-hop HoVer claim verification dataset and found that it greatly outperforms the baseline models.

## Acknowledgments and Disclosure of Funding

We would like to thank Ashwin Paranjape, Megha Srivastava, and Ethan A. Chi for valuable discussions and feedback. This research was supported in part by affiliate members and other supporters of the Stanford DAWN project—Ant Financial, Facebook, Google, and VMware—as well as Cisco, SAP, a Stanford HAI Cloud Credit grant from AWS, and the NSF under CAREER grant CNS-1651570. Any opinions, findings, and conclusions or recommendations expressed in this material are those of the authors and do not necessarily reflect the views of the National Science Foundation.

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
