## Appendices for Baleen

## A    Data Details

Table 6: Sizes of the splits of the datasets used in this work.

| Multi-Hop Dataset | Train | Dev | Test |
|---|---|---|---|
| HotPotQA | 90,447 | 7,405 | 7,405 |
| HoVer | 18,171 | 4,000 | 4,000 |

As our retrieval corpus for both HotPotQA and HoVer, we use the Wikipedia dump released by Yang et al. [31] from Oct 2017,[4] which is the Wikipedia dump used officially for both datasets. For each Wikipedia page, this corpus contains only the first paragraph and the passages are already divided into individual sentences. It contains approximately 5M passages (1.5 GiB uncompressed). We use the official data splits for both datasets, described in Table 6.

## B    Baleen implementation & hyperparameters

We implement Baleen using Python 3.7 and PyTorch 1.6 and rely extensively on the HuggingFace Transformers library [28].[5] We train and test with automatic mixed precision that is built into PyTorch.

### B.1    FLIPR retriever

Our implementation of FLIPR is an extension of the ColBERT [16] open-source code,[6] where we primarily modify the retrieval modeling components (e.g., adding focused late interaction and including condensed fact tokens in the query encoder).

For FLIPR, we fine-tune a BERT-base model (110M parameters). For each round of training, we initialize the model parameters from a ColBERT model previously trained on the MS MARCO Passage Ranking task [20]. To train the single-hop retriever used to initiate the supervision procedure of §3.2, we follow the training strategy of Khattab et al. [17]. In particular, we use this *out-of-domain* ColBERT model to create training triples, and then we train our retriever (in this case, FLIPR for first-hop) with these triples. Once we have this first-hop model, the rest of the procedure follows Algorithm 1 for latent hop ordering.

Table 7 describes our hyperparameters for FLIPR. We manually explored a limited space of hyperparameters in preliminary experiments, tuning validation-set Retrieval@$k$ Accuracy, with $k$=100 for HoVer and $k$=20 for HotPotQA, while also being cognizant of downstream Psg-EM and Sent-EM. We expect that a larger tuning budget would lead to further gains, which is consistent with the fact that our Psg-EM and Sent-EM on the *held-out* leaderboard test set are 1.0 and 0.6 points *higher* than the public validation set with which we developed our methods. Unless otherwise stated, the hyperparameters apply to both FLIPR and ColBERT.

We adopt the default learning rate from ColBERT, namely $3 \times 10^{-6}$. We set the embedding dimension to the default $d = 128$ and use a batch size of 48 triples. We truncate passages to 256 tokens. For the query encoder, we truncate queries to 64 tokens and allow up to 512 tokens in total, particularly for the condensed facts or reranker context passages from the previous hops. We adopt the ColBERT infrastructure for end-to-end retrieval, where candidate generation is done using nearest-neighbor search with FAISS [14].[7] See Khattab and Zaharia [16] for use with late interaction models. Since multi-hop queries are much longer than standard IR or OpenQA queries, we conduct candidate generation for FLIPR and ColBERT using the embeddings corresponding to the query tokens (defined in §3.1), and then apply (focused) late interaction on both the query and facts embeddings.

The settings for LHO are also summarized in Table 7, namely the depth (i.e., number of passages) used for sampling negative passages and positive passages and for extracting multi-hop context facts.

---

[4] https://hotpotqa.github.io/wiki-readme.html
[5] https://github.com/huggingface/transformers
[6] https://github.com/stanford-futuredata/ColBERT
[7] https://github.com/facebookresearch/faiss/

Table 7: Hyperparameters for Baleen's FLIPR on HoVer and HotPotQA.

| Hyperparameter | HoVer | HotPotQA |
|---|---|---|
| Embedding Dimension | 128 | 128 |
| Maximum Passage Length | 256 | 256 |
| Maximum Query Length: query/overall | 64/512 | 64/512 |
| Focused Interaction Filter: $\hat{N}$ / $\hat{L}$ | 32/8 | 32/8 |
| Learning rate | $3 \times 10^{-6}$ | $3 \times 10^{-6}$ |
| Batch size (triples) | 48 | 48 |
| Training steps (round #1; per hop) | 10k, 5k, 5k, 5k | 20k, 20k |
| Training steps (round #2) | 10k | 40k |
| Training Negative Sampling Depth (for each hop) | 1000 | 1000 |
| Training Context (Fact) Extraction Depth (from each hop) | 5 | 5 |
| Training Positive Sampling Depth (round #1; per hop) | 20, all, all, all | 20, all |
| Training Positive Sampling Depth (round #2; per hop) | 10, 10, 10, all | 10, all |
| FAISS centroids (probed) | 8192 (16) | 8192 (16) |
| FAISS results per vector: training/inference | 256/512 | 256/512 |
| Inference Top-$k$ Passages Per Hop | 25, 25, 25, 25 | 10, 40 |

## B.2 Condensers and rerankers

For the two-stage condenser, we train two ELECTRA-large models (335M parameters each), one per stage. We simply use a [MASK] token to separate the facts, although we expect that any other special-token choice would work similarly provided enough training data is available.

We train the first-stage condenser with ⟨query, positive passage, negative passage⟩ triples. We use a cross-entropy loss over the individual *sentences* of both passages in each example, where the model has to select the positive sentence out of ⟨$positive_j$, $negative_1$, $negative_2$, ...⟩ for each positive $j$. We average the cross-entropy loss per example, then across examples per batch. We train the second-hop condenser over a set of 7–9 facts, some positive and others (sampled) negative, using a linear combination of cross-entropy loss for each positive fact (against all negatives) and binary cross-entropy loss for each individual fact.

Table 8: Hyperparameters for Baleen's condensers on HoVer and HotPotQA.

| Hyperparameter | HoVer | HotPotQA |
|---|---|---|
| Learning rate | $1 \times 10^{-5}$ | $1 \times 10^{-5}$ |
| Batch size | 64 | 64 |
| Maximum Sequence Length | 512 | 512 |
| Warmup Steps | 1000 | 1000 |
| Training steps (stage #1) | 5k | 10k |
| Training steps (stage #2) | 5k | 10k |
| Negative Sampling Depth (stage #1; per hop) | 20, 20, 20, 20 | 10, 30 |
| Negative Sampling Depth (stage #2; for each hop) | 10 | 10 |
| Context Sampling Depth (from each hop) | 5 | 5 |
| Positive Sampling Depth (stage #1; per hop) | 10, 10, 10, all | 10, all |
| Positive Sampling Depth (stage #2; for each hop) | 10 | 10 |
| Facts fed to stage # 2: training (inference) | 7–9 (9) | 7–9 (9) |

Table 8 describes our hyperparameters for the condensers.

**Claim verification** For HoVer, we train an ELECTRA-large model for claim verification. The input contains the query and the condensed facts and the output is binary, namely supported or unsupported. We use batches of 16 examples and train for 20,000 steps with 2–6 facts per input, but otherwise adopt the same hyperparameters as the second-stage condensers.

**Reranker** We similarly use ELECTRA-large for the rerankers. The input contains the query and one reranker-selected passage for each of the previous hops as well as one passage to consider for the current hop. We adopt the same positive and negative sampling as the first-stage condenser, as well as

other hyperparameters, but we allow twice the training budget when tuning the number of steps since there is only one stage for reranking. During training and inference, a primary difference between the reranking and condensing architectures is that condensing takes zero or more new facts per hop, whereas reranking always adds exactly one per hop as additional context, namely the top-ranked passage (positive) for inference (training).

**Hybrid condenser/reranker implementation** We train a single retriever on top of the first-round retrievers for the condenser and reranker Baleen architectures. During inference, we run two independent pipelines, one with the condenser and the other with the reranker. After both retrieval pipelines have completed, we merge the overall top-100 results by taking the top-13 and top-12 per hop from both pipelines without duplicates, for a total of $25 \times 4 = 100$ unique passages.

## B.3 Heuristic ordering: using overlap between passages and titles

To ablate our latent hop ordering (LHO) approach, we consider an algorithm that generalizes Xiong et al. [29]'s heuristic for ordering the two HotPotQA passages for training. This heuristic applies to any number of hops and attempts to capitalize on our understanding of the construction of HotPotQA and HoVer: each hop's passage is referred to directly by at least one other passage, often by title.

Where short answer strings are available (i.e., in HotPotQA), if only one passage contains that answer string, we treat it as the final hop's passage and consider the order among the remaining passage(s). We then assign a score to all remaining passages, which reflects whether the full or partial title of the passage appears inside the claim. We take the highest-scoring passage(s) as the target of the current hop, expand the claim with their full text, and repeat this process with any remaining passages.

In case all passage titles are completely unmatched, which is relatively uncommon, we recurse by attempting to pick each possible candidate passage for the current hop, and keep the choice that (at the end of the recursion) results in the largest overlap scores for the remaining hops. For the ablations that use this heuristic, we train in two rounds like with LHO. However, the first round with this heuristic does not require multiple short training runs, one per hop, since all the hops are determined in advance. Instead, we allow the same total training budget for the first round (as well as the second) over all hops.

We note that an additional signal for the order of hops is to recover (from Wikipedia) the hyperlinks between the passages and use them to build a directed graph. However, this violates our assumption that we are only given the text of the passages and would make very strong assumptions about the data, ones specific to the construction of HotPotQA and HoVer.

## B.4 Resources used

We conducted our experiments primarily using internal cluster resources. We use four 12GB Titan V GPUs for retrievers and four 32GB V100 GPUs for condensers, rerankers, and readers. Training FLIPR on the four-hop HoVer dataset requires five (4+1) short training runs, for a *total* time of approximately five hours. Similarly, we encode and index the corpus five times in total (four intermediate and one final time) less than six hours in total. Retrieving positives and negatives for training from the index four times consumes a total of less than three hours. All four hops of retrieval on the validation set with the final FLIPR model take a total of a little over one hour. Training both condenser stages for 5k steps each and training the claim verification reader for 20k steps takes a total of less than eight hours. We use python scripts for pre- and post-processing (e.g., for LHO) and run the condensers during evaluation, which generally add only limited overhead to the running time of our experiments.

Our FLIPR retriever adopts a fine-grained late interaction paradigm like ColBERT (see §2), so our memory footprint is relatively large, as it involves storing a small 256-byte (2-byte 128 dimensions) vector per token. The uncompressed index is about 83 GiBs. We note that the authors of ColBERT Khattab and Zaharia [16] have recently released a quantized implementation that can reduce the storage per vector 4–8 fold, and reducing the storage space of dense retrieval methods through compression and quantization while preserving accuracy is an active area of research [12; 30], with recent encouraging results.

## C   The effect of condensing on the context lengths

We compare our condenser architecture of Baleen to a reranker after four hops on HoVer. We find that the average context per query is 91 words for Baleen's condenser architecture versus 325 words for the reranking ablation, on average. This $3.6\times$ improvement for Baleen's condenser suggests that for tasks with even more hops, a condenser approach would be less likely to overwhelm typical maximum sequence lengths of existing Transformer architectures.

By design, the condenser architecture scales better to a large number of hops than existing work. For instance, MDR uses beam search, and would require exploring a very large set of paths (e.g., $100 \times 100$) from the third hop onwards. As another example, IRRR concatenates long passages together as it conducts more hops. In contrast, the condenser architecture only needs to concatenate short facts together. While this efficiency gap grows when more than two hops are necessary, condensing maintains a comparable cost to MDR and IRRR in the simple two-hop (HotPotQA) setting, where Baleen invokes an ELECTRA-large model only tens of times per query, namely for the top-10 passages from the first hop, the top-40 passages from the second hop, and twice in the second stage of condensing.