# OpenReview forum: "Baleen: Robust Multi-Hop Reasoning at Scale via Condensed Retrieval"
_NeurIPS.cc/2021/Conference — NeurIPS 2021 Spotlight_

### Official Review · Reviewer_EkjX · 2021-07-13

**Rating:** 8
**Confidence:** 5

**Summary:**

**Summary**: This paper presents a new retrieval system for multi-hop QA, i.e., answering the questions requires multiple relevant documents. On top of existing dense retrieval techniques, this paper proposes three independent techniques to tackle multi-hop retrieval problems, including a topk multi-vector scoring function, condensed fact selection for query reformulation and a latent method to recover the proper retrieval order during training. This system achieves significant improvements on two recent multi-hop datasets.

**Contribution**:
1. A strong multi-hop retrieval system that outperforms existing systems by a large margin.
2. This is the first work showing that dense retrieval can be generalized to more than 2 hops.
3. Thorough experiment results and ablations.

**Limitations And Societal Impact:**

I don't see any negative impact of this work.

**Main Review:**

Originality: This is a new system with a combinations of some existing techniques or their modifications. For example, the latent hopping order is already explored by IRRR (Qi et al.) However, I think the overall system is valuable given the nice work of combining the best practice and getting significant improvements. The related work is adequately discussed.

Quality: The experiments are solid and the proposed methods are reasonable. The authors are also very clear about the limitations of their work. For instance, the proposed system requires sentence-level evidence annotations.

Clarity: Overall, the paper is well-written and easy to follow. For further improvements, the authors could provide more details about the implementations. For example, how many sentence-level facts are appended to the current query and other hyperparameters.

Significance: The results of this paper are important and valuable. It has pushed the retrieval performance of HotpotQA. Any end-task models should be able to reuse these results.



**Time Spent Reviewing:**

2

---

> ### Author Response · Authors · 2021-08-09
> **Response to Reviewer EkjX**
>
> We thank you for the positive review of our work. We are thrilled that you found our experiments and ablations thorough and found our system strong, noting that it shows dense retrieval generalizing to more than two hops for the first time.
>
>
> > **the latent hopping order is already explored by IRRR (Qi et al.) However, I think the overall system is valuable given the nice work of combining the best practice and getting significant improvements. The related work is adequately discussed.**
>
> We are keen to further clarify the connection to IRRR in terms of supervision if our work is accepted (using the extra page).
>
> As we note in L134, “our supervision for the first hop shares inspiration with GoldEn [17], which uses a bag-of-words model to identify the ‘more easily retrieved’ passage among just two for HotPotQA.” Indeed, IRRR also employs similar methods to GoldEn, as the IRRR paper notes in Sec 3.4.1. However, we believe that there are crucial differences to Baleen: to the best of our understanding, both GoldEn and IRRR extract term-based search queries (and the order) from _term overlap_ between the reasoning path and the target paragraph and, incidentally, focus on training in the _two-hop_ setting (HotPotQA). We represent this (as well as MDR’s approach) with the term-overlap heuristic for ordering that is employed in our ablation study. In contrast to existing work, Baleen's LHO method uses the learned/trained retriever itself to find strong orderings and targets the many-hop training case, which is harder due to the presence of intermediate passages between the first and the last.
>
>
> > **For further improvements, the authors could provide more details about the implementations. For example, how many sentence-level facts are appended to the current query and other hyperparameters.**
>
>
> Thank you for the feedback. We will expand on the main text to address more implementation details if this work is accepted. To answer your question, we simply append the facts whose score is positive (i.e., exceeds threshold=0), but in the rare case that we have more than five sentences (hyperparameter), we drop the lower-scoring ones.

---

### Official Review · Reviewer_buZr · 2021-07-16

**Rating:** 7
**Confidence:** 3

**Summary:**

This paper introduces a new multi-hop reasoning system called Baleen which conducts iterative retrieval and condensing. Baleen repeats the iterative process for pre-defined time steps T (e.g., T=2 for HotpotQA and T=4 for HoVer)  and at every time step, it takes the query from the previous time step and retriever top K passages, and create a new query for the current t step by extracting and condensing the facts (sentences) from the retrieved passages. When they finish retrieval (i.e., t=T), they feed the final query to the reader model to get the final predictions.
They conducted experiments on HotpotQA and HoVer. The experimental results on HotpotQA show its competitive passage retrieval performance and they show state-of-the-art results on HoVer, significantly outperforming the baselines.


**Ethical Concerns:**

I don't think the paper has any major ethical issues.

**Limitations And Societal Impact:**

I appreciate that the authors provide detailed discussions on the limitations and social impacts and I generally agree with the points discussed there and I enjoy reading them. Regarding the hallucinations by giant generative models, some recent retriever-generator frameworks such as Fusion-in-Decoder (Izacard and Grave, 2020) provide attention scores which help us to understand which passages are looked at by the model during answer generation, so I’m not sure if it’s often more ungrounded compared to retriever-extractor frameworks.

**Main Review:**

**Originality.** The multi-hop retrieval and reasoning has been widely studied and considered a challenging task. Although I think the proposed approach itself is solid and technically sound, more discussions on the prior multi-hop retrieval work would be helpful. Several prior work augments search queries for multiple steps with retrieved segments from the context e.g., IRRR (Qi et al., 2020) or GOLDEN (Qi et al., 2021). Another direction is to reformulate the queries in *a latent space* (Multi-step retriever-reasoner by Das et al., 2019 or MUPPET by Feldman and El-Taniv, 2019), and they should be mentioned in the paper as well.

**Quality.** Overall, I think the proposed approach is technically sound and well-motivated. The experimental results show its effectiveness on two multi-hop reasoning datasets. Yet, I have two concerns and questions. First, Baleen assumes that the latent ordering hop setting, where we have gold paragraph annotations but lack the annotation on its order. I think this setting relies on the two target datasets, HotpotQA and HoVer; many QA datasets come without gold paragraph annotations (e.g., TriviaQA), and relying on the existence of the gold paragraph annotations may limit the framework’s applicability to a wider set of datasets. Do you think your approach can work if we do not have gold passage annotations (e.g., heuristically decide gold passages using entity linking or something)? The second concern is on evaluation and comparison with state-of-the-art multi-hop retrievers. Although I understand the main focus of the paper isn’t on improving reader components, I would like to see at least the official supporting fact retrieval performance on HotpotQA, not only unofficial paragraph retrieval scores. The HoVer baseline (TF-IDF + BERT) is a bit weak. Is it possible to test some more recent and strong baselines like SemanticRetrievalMRS (Nie et al., 2019, https://github.com/easonnie/semanticRetrievalMRS) on HoVer? I think they study on both HotpotQA and FEVER so the models and codes should be applied to HoVer as well.

**Clarity.**
I think the paper is well written and easy to read.

**Significance.**
This paper introduces a new approach for multi-hop retrieval and is probably useful for future work in this domain. Although it advances state-of-the-art results on two datasets, I wish the evaluation results with official evaluation metrics on HotpotQA were presented. The ablation results might also provide useful insights. Heuristically constructing reasoning orders or chains have been widely used, but they show that it doesn’t perform well on complex questions.


**Time Spent Reviewing:**

2

---

> ### Author Response · Authors · 2021-08-09
> **Response to Reviewer buZr**
>
> We thank you for the thorough review of our work! We are glad that you view our approach as solid, technically sound, and well-motivated and that you find the paper well-written and easy to read.
>
>
> > **Another direction is to reformulate the queries in a latent space (Multi-step retriever-reasoner by Das et al., 2019 or MUPPET by Feldman and El-Taniv, 2019)**
>
> Thank you for raising this point. If this work is accepted, we are eager to use (part of) the extra page to discuss these important papers as well.
>
>
> > **Q: Do you think your approach can work if we do not have gold passage annotations (e.g., heuristically decide gold passages using entity linking or something)?**
>
> Thank you for highlighting this: we agree that supervision without any passage labels is an important and exciting problem, and as you say a combination of entity linking and zero-shot retrieval might support progress on this front.
>
> Baleen will not directly work without passage annotations, though LHO makes Baleen effective without an explicit ordering of the gold passages. In general, passage-less supervision appears easier for QA (e.g., HotPotQA) than for claim verification (e.g., HoVer), because QA can rely on answer overlap as a signal whereas claim verification has no clear signal. In that case, zero-shot retrievers can generally resolve the first hop, and if two-hop “answers” are available, it might be possible to learn the “extra hop” toward passages that contain the answer string, then build on this for the “third” hop onwards.
>
> We believe this is an exciting direction for work that builds directly on Baleen, but we do not currently view this as part of our scope.
>
>
> > **Q: Although I understand the main focus of the paper isn’t on improving reader components, I would like to see at least the official supporting fact retrieval performance on HotpotQA**
>
> As you observe, we focus on retrieval in this work, tackling the three challenges outlined in the introduction. Large gains in retrieval can translate to great gains on the downstream tasks (as we show on HoVer), but we find that existing systems have extremely high recall values on HotPotQA already, making the competition on downstream metrics largely reader-driven. This naturally gives an advantage to more elaborate reader design and reader training (e.g., the IRRR authors train on both SQuAD and HotPotQA, doubling the amount of training data), which lie outside our scope.
>
> However, we did in fact apply a quick test in response to your comment and we are eager to hear your feedback. Using the MDR code, we plugged in the MDR reader as-is on top of our retrieval results. We want to emphasize that this was an **off-the-shelf reader**, without any adaptation or fine-tuning to work with our retrieval. Below is the summary of the official (non-retrieval) metrics of HotPotQA results on the dev set.
>
> - Ans-EM is raised by +3.5 to 65.8 from 62.3.
> - Ans-F1 is raised by +3.6 to 78.6 from 75.0.
> - Sp-EM is raised by +2.9 to 59.4 from 56.5.
> - Sp-F1 is raised by +4.4 to 83.8 from 79.4.
> - Joint-EM is raised by +2.2 to 44.3 from 42.1.
> - Joint-F1 is raised by +3.6 to 69.9 from 66.3.
>
> We think of this as a **lower bound** on our downstream improvements, _as we simply used an off-the-shelf reader from MDR without adaptation to Baleen_. These gains are consistent with the expectation of **reviewer EkjX** that various end-task models can re-use the stronger retrieval.
>
>
> > **The HoVer baseline (TF-IDF + BERT) is a bit weak. Is it possible to test some more recent and strong baselines like SemanticRetrievalMRS (Nie et al., 2019) on HoVer?**
>
>
> We agree with you that the default HoVer baseline TF-IDF + BERT is underwhelming indeed. This motivated our choice of **ColBERT-Hop** for additional comparison, which we find to be a very strong baseline as explained below.
>
> We thank you for suggesting SemanticRetrievalMRS. This work has been influential in this area, but considering its publication in 2019, we worry that it is no longer competitive in this fast-moving space. For instance, on HotPotQA, the passage-pair EM (P-EM) of Semantic Retriever appears to be 63.9 (figure from the MDR paper), compared with MDR’s 81.2, IRRR’s 84.1, ColBERT-Hop’s 85.2, and Baleen’s 86.7. It also appears that Semantic Retriever is 17 points lower in answer-match quality against MDR. We also appreciate that the authors of SemanticRetrievalMRS have made their code available. However, considering the very fast pace in this area and that their code has not been updated in two years, we worry that it might be challenging to adapt and run (especially on new datasets with more than two hops and with the latest infrastructure and libraries like CUDA, Huggingface, etc.), which would reduce our confidence in the validity of comparisons.
>
> To alleviate the weaknesses of the HoVer official baseline, as we explain in the paper, we use (as baseline) a many-hop flavor of ColBERT that adopts the IRRR greedy-selection pipeline, dubbed **ColBERT-Hop**. For completeness, let us reiterate ColBERT-Hop’s design: it uses the state-of-the-art dense retriever ColBERT from IR and Open-QA (shown to greatly outperform term-based search like IRRR’s and single-vector search like MDR’s), uses greedy passage search like IRRR and other popular systems, and uses overlap-based ordering for training similar to MDR.
>
> We use this as a baseline on HoVer given that: (a) ColBERT-Hop outperforms both MDR and IRRR on HotPotQA retrieval, (b) MDR code doesn’t currently handle more than two hops and the IRRR code is not publicly available so we can’t use them on HoVer, and (c) ColBERT-Hop is capable of scaling in training and inference to many hops and, by itself, demonstrates impressive results on both HotPotQA and HoVer.

---

> ### Comment · Reviewer_buZr · 2021-08-24
> **Thank you for your response**
>
> Thank you for your detailed response and additional results. I think my main concerns are mostly addressed by them, and therefore I raised my score to 7 (accept).

---

### Official Review · Reviewer_EV88 · 2021-07-17

**Rating:** 7
**Confidence:** 4

**Summary:**

This paper proposed Baleen a system that improves the accuracy of multi-hop retrieval while learning robustly from weak training signals in the many-hop setting. Baleen generally contains three modules -- Focused Late Interaction, Latent Hop Ordering, and Condenser. Focused Late Interaction is an improved version of ColBERT, in which they only consider $\text{top-}\hat{K}$ partial scores instead of all scores. Latent Hop Ordering proposed a way to supervise the retriever using weak supervision per-hop. The condenser summaries the sentence-level information retrieved per-hop so it could be better used for the next hop. Experimental results on two multi-hop-based datasets, namely HotpotQA and HoVer show that the proposed framework is quite effective for retrieving multi-hop evidence.

**Limitations And Societal Impact:**

The authors have sufficiently addressed the limitations of their work and I don't think there is a potential negative societal impact.

**Main Review:**

Strength:
1. The authors implemented a multi-hop retrieval system that achieved very strong performance on two multi-hop-based datasets.
2. The idea behind the proposed modules is simple yet effective.
3. Reasonable ablations showing the effectiveness of each module.

Weaknesses:
Although the system proposed in this paper is quite strong, there are not many insights I can draw from the paper as many of the proposed architecture changes are heuristic-based, e.g., taking $\text{top-}\hat{K}$ partial scores is straightforward but how do we choose $\hat{K}$? The idea of using a condenser is also quite intuitive, but how do we choose the negatives and how many negatives are needed? As we can see in the appendix, there are so many hyperparameters in the appendix and I am not sure to what extent those hyperparameters would affect the performance.

Question for the authors:
1. Compared to some of the previous literature, e.g., IRRR, DPR, the proposed framework seems much more computationally expensive; do you have an idea of how slow the proposed method is compared to those models for both training and inference?


**Time Spent Reviewing:**

4-5

---

> ### Author Response · Authors · 2021-08-09
> **Response to Reviewer EV88**
>
> We thank you for the valuable review of our work. We are glad that you found our results very strong and the ablations we used to show the effectiveness of each module reasonable.
>
>
> > **Although the system proposed in this paper is quite strong, there are not many insights I can draw from the paper as many of the proposed architecture changes are heuristic-based.**
>
> Thank you for the feedback. If this work is accepted, we will use the extra space to tie each of our contributions back to the big picture, as below.
>
> (1) With more than two hops, the search space grows extremely large (e.g., MDR's search with top-100 passages per hop would yield 100M paths after four hops). Breaking out from the typical design space in this area, we suggest condensed retrieval to summarize/extract all the pertinent facts from each hop. This yields a shorter context than even greedy passage selection while achieving very high recall (see appendix C and table 5).
>
> (2) With complex queries, we must be able to retrieve documents that match only *part* of the query but we cannot always know which part in advance. IRRR and GoldEn filter the retrieved passages into fewer terms *before* search, but this would fail in some important cases. We highlight one such example in the paper, where the query contains “World Series 1955, 1958, 1965, and 1970” and talks about an MVP but the model should select the document for “World Series 1965”, which is the only one that also refers to a prominent “MVP”, a fact only seen *after* finding the document. We introduce “focused late interaction” which allows each document to pay attention to a different part of the same query representation _without compromising the scalability of late interaction_. In doing so, we are the first to show that scalable late interaction can be modified to increase its expressive power without compromising its efficiency. Moreover, the suggested top-k matches mechanism applies a _post-interaction_ filter, which allows the full query to see the full document and yet mitigates penalizing the document for not matching irrelevant parts of the complex query.
>
> (3) With more than two hops, supervision can be difficult without knowing the order of the hops. We suggest a robust mechanism where the retriever itself selects the supervision, which we hope (as Reviewer buZr notes under “Significance”) might have significant value beyond our work as existing literature has relied on dataset-dependent heuristics for this.
>
>
> > **Taking $\hat{K}$ partial scores is straightforward but how do we choose $\hat{K}$? The idea of using a condenser is also quite intuitive, but how do we choose the negatives and how many negatives are needed? As we can see in the appendix, there are so many hyperparameters in the appendix and I am not sure to what extent those hyperparameters would affect the performance.**
>
>
> In the appendix, we report as many hyperparameters as possible (though we did forget one, clarified below) in the spirit of reproducibility and transparency. That said, many of these hyperparameters are very standard ones. For instance, most of the generic parameters for the retriever (e.g., embedding dimension, negative sampling depth, learning rate) are borrowed defaults from the ColBERT system which we extend to build FLIPR (see L480 in the appendix). We will revise the paper to make this clearer.
>
> Thank you for noting the partial scores $\hat{K}$ (or $\hat{N}$ for consistency with the paper); we have indeed missed this hyperparameter. For the query, we simply set $\hat{N}$ = $N$/2 (i.e., half of the query embeddings, which maximizes the number of subsets that can be selected or dropped). For the extracted facts, we set $\hat{N}$=8. We use those as simple default choices for both HoVer and HotPotQA, though initial preliminary experiments suggested other values work well too. We will clarify this in revising our work.
>
> To train the first stage of the condenser, we just need triples (one negative, one positive per query) and we sample hard negatives using the FLIPR checkpoint used for indexing the corpus. The depth of sampling the negative is kept as the simple default 1000, just like many other retrieval systems. To train the second stage, we found that simply selecting 7--9 sentences that include all the positives (and negatives from hard-negative documents) suffices. We choose the 7--9 size as the largest number of sentences that is likely to fit within the 512-token limit of common transformer models in practice. We allow for some variation (7,8,9) to ensure the model is robust to different input lengths. We will make sure our implementation is clearer in the final version, if our work is accepted.
>
>
>
>
> > **Q: The proposed framework seems much more computationally expensive; do you have an idea of how slow the proposed method is compared to those models for both training and inference?**
>
> We thank you for raising this important point about efficiency. If our work is accepted, we will expand on this important dimension (as below). By design, our framework is meant to be _more_ scalable to more hops than existing work. Whereas MDR uses beam search and IRRR concatenates long passages together, we only need to concatenate short facts together. See appendix C for the effect of condensing on the context lengths.
>
> While Baleen’s relative efficiency shines with more hops, it has a comparable cost to MDR and IRRR in the simple two-hop (HotPotQA) setting. Below, we sketch the relative costs of Baleen, MDR, and IRRR in terms of invocations to the transformer models.
>
> - Baleen: The query is fed to FLIPR (BERT-base) once. The top-10 passages are fed to the condenser (ELECTRA-large). The query is extended with extracted facts and is fed to FLIPR (BERT-base) again. The new top-10 passages are once more fed to the condenser (ELECTRA-large). Overall, we invoke BERT-base on two sequences and ELECTRA-large on 20 sequences for retrieval and condensing.
>
> - MDR: The query is fed to a DPR retriever (BERT-base) once. The top-100 passages (plus the original query) are then fed to BERT-base again for the second hop. MDR selects the top-100 overall pairs of passages. We might also want to consider the re-ranking stage, where MDR re-ranks the top-100 pairs with ELECTRA-large. Overall, MDR invokes BERT-base for 101 sequences and ELECTRA-large on 100 sequences for retrieval and re-ranking. The key point, however, is that scaling MDR to just one more hop would explode because beam search would consider potentially 100x100 paths as input to the third hop. This beam size can be reduced, but this could lower the recall of MDR considerably. We thus argue that Baleen outperforms MDR in both recall *and* scalability.
>
> - IRRR: The query is fed to ELECTRA-large once. The top-100 passages are re-ranked with ELECTRA-large. The top-ranked passage is then appended to the query, and the query is again fed to ELECTRA-large. The top-100 passages are again re-ranked with ELECTRA-large. Even though HotPotQA is a two-hop task, IRRR has a selection mechanism that might force it to do three, four, or even five hops to find the two passages. Overall, IRRR needs at least 201 invocations (at most 501 invocations) to ELECTRA-large for retrieval.
>
>
> Another dimension is that Baleen uses “late interaction” with (128-dim) multi-vector representations instead of MDR’s (768-dim) single-vector representations, but late interaction uses scalable approximate nearest neighbor search and so is efficient in its own right, though single-vector retrieval is certainly even cheaper. (Late interaction involves a matrix multiplication instead of a dot product, for each of the documents retrieved with approximate nearest-neighbor search. In general, these matrix multiplications are cheap on GPUs and are dominated in cost by invocations to large transformer models.) Similarly, we do believe that our training scheme has a similar cost to related work, though we do not have a direct comparison at the moment. For the four-hop HoVer task (see L556 in appendix), training our four-hop FLIPR retriever (on a single machine with four Titan V GPUs) takes just five hours plus less than six hours for [re-]encoding the corpus accordingly.

---

> > ### Author Response · Authors · 2021-08-28
> > **Response to Reviewer EV88**
> >
> > We would like to make a small correction to our response to reviewer **EV88**. On HotPotQA, Baleen requires 52 ELECTRA invocations. Our response to reviewer **EV88** originally accounted for 20 invocations only (i.e., 10+10 for the two hops, which was our original implementation, instead of currently 10 for the first hop, +40 for the second hop, and +2 for the second-stage condenser). We will update our earlier response above after a few days to reflect this; we keep it unchanged now for reference.
> >
> > This does not affect our earlier response's argument, namely that _"While Baleen’s relative efficiency shines with more hops, it has a comparable cost to MDR and IRRR in the simple two-hop (HotPotQA) setting."_ In particular, Baleen's cost on HotPotQA is in line with the 100 invocations for MDR and the >= 201 invocations for IRRR. To reiterate an important point, Baleen's design excels at scaling to more hops: for the 4-hop HoVer dataset, it just condenses 25x4=100 passages, whereas beam search might consider far more candidates (e.g., 100^3 or 25^3 sequences out of the first three hops) and standard greedy search requires considerably longer sequences (appendix C).
> >
> > **None of this affects the content in the paper** — the paper as-is correctly reflects this (see Table 7 for the number of passages per hop on HoVer and HotPotQA). We will make sure to make these efficiency comparisons with MDR/IRRR clear in our revision, if this work is accepted.

---

### Official Review · Reviewer_ZzDs · 2021-07-19

**Rating:** 8
**Confidence:** 5

**Summary:**

The paper makes contributions to passage retrieval in the open-domain, multi-hop setting.  It builds on recent work which applied dense passage retrieval iteratively to work for multi-hop tasks (such as MDR and IRRR).  It proposes three modifications to existing work: condensed retrieval, which summarizes retrieved passages at each hop to reduce the search space; focused late interaction, which is a variation on Colbert to consider only top-k scores for each query embedding; and latent hop ordering, which is a learned strategy to order the passages which improves over existing heuristics.  Each of these modifications are relatively minor, but together lead to very strong results on the often used HotpotQA and the newly released HoVer benchmarks.

**Limitations And Societal Impact:**

Yes

**Main Review:**

The paper builds on recent advances in multi-hop retrieval using dense representations.  The contributions are not ground-breaking, but clear.  The ideas are well executed, with strong results and clearly presented.  Overall this is a solid candidate for publication at NeurIPS.

Originality:  The contributions consist of 3 improvements to multi-hop dense retrieval.  2 of these (focused late interaction and latent hop ordering) are very minor and can be considered incremental.  The remaining one, condensed retrieval, is a neat and novel idea which incorporates summarization into multi-hop retrieval.  All three contributions are clear, well executed and clearly demonstrated to improve performance.  The end result is a much better, 'polished' system for multi-hop dense retrieval.

Significance:  Unfortunately 'multi-hop reasoning' (despite the exciting name) suffers from a lack of natural benchmarks and real-world applications, which somewhat limits the significance of advances in this area.  Nevertheless, this is not a weakness of the paper.  The results are strong, and the methods are clear and simple enough for others to adopt.

Quality:  The experiments are conducted against very strong baselines, and the improvements are convincing.  There are enough ablations to demonstrate the contribution of each component.

Clarity:  The presentation is clear and generally easy to follow.  The authors make it very clear what the current SotA is and exactly what their contributions are.

Questions/feedback:
- What is the criteria for claiming "saturation" for HotpotQA?  Given no human upper-bound, it would be helpful to make this section more formal.  While many are eager to 'retire' HotpotQA, I don't think simply attaining 90+ accuracy is sufficient to make the claim.  One suggestion could be to look at the remaining examples, and show they are somehow irrelevant (e.g. bad annotation, ambiguous question etc.)
- What does top-20 passage recall mean exactly when condensing passages?
- Some of the important implementation / training details can be moved to the main text from the appendix.  Currently the manuscript is pretty light in details (if one doesn't read the appendix).

**Time Spent Reviewing:**

1.5

---

> ### Author Response · Authors · 2021-08-09
> **Response to Reviewer ZzDs**
>
> We thank you for the thoughtful feedback. We are glad you found condensed retrieval and the overall system strong, well-executed, and well-presented. We also appreciate that you found our baselines very strong and our ablations convincing.
>
> > **Q: What is the criteria for claiming "saturation" for HotpotQA? Given no human upper-bound, it would be helpful to make this section more formal.**
>
> Thank you for raising this important question. If our work is accepted, we will make use of the additional page toward making our claim clearer and narrower. To begin with, our comment is focused on top-k retrieval. Other aspects of the dataset are still challenging and important, especially answer extraction with challenging distractors.
>
> Concretely, we observe that, when it comes to top-k retrieval, existing systems are very close to reducing HotPotQA’s “fullwiki” setting to a “distractor” setting, where a small set of passages is extremely likely to contain the answer.  Downstream “reader” or re-ranker models can generally read 20--100 passages, and we find that Baleen’s answer-recall@20 is 96.3% (pair-recall@20 is 93.3%) and its answer-recall@100 is over 98% (pair-recall@100 is almost 97%). For comparison: to our knowledge, on NaturalQuestions and TriviaQA, the highest answer-recall@20 is approximately 85% (by ColBERT-QA; TACL’21), even though NQ/TQ are single-hop datasets, and thus presumably (much) easier.
>
> We will revise the paper to clarify the following: We want to highlight that, when comparing very high-recall systems like Baleen, our own baseline ColBERT-Hop, IRRR, and even MDR, the competition in downstream quality will naturally be reduced to a large extent to the reader, since it almost always has access to the answer in the top-k set and is thus responsible for the majority of downstream failures. This does not necessarily imply that additional retrieval gains are not possible: for all we know, it might still be possible to increase answer-recall@20 to 99% from Baleen’s 96.3%. Indeed, we sampled five random examples where Baleen fails to find one or both gold passages in the top 20, and only two stood out as straight-out “unsolvable” (at least without assuming hyperlinks).
>
>
> > **Q: What does top-20 passage recall mean exactly when condensing passages?**
>
> Thank you. We will clarify the following in the revision if the work is accepted.
>
> The paper defines this as “the percentage of questions [...] for which the model’s top-k retrieved passages contain both gold passages”. While condensed retrieval selects only a few facts from each hop, we can still keep track of the top passages from each hop’s retrieval. In the case of HotPotQA, Baleen’s top-20 passages are simply the top-10 passages from the first hop merged [excluding duplicates] with the top-10 passages from the second hop. (Similarly for HoVer, the top-100 passages are the top-25 per hop; see L268).
>
> Besides the condensed context for hopping between passages, this “top-k” set is useful for comparison with other systems, for showing provenance or sources to users (e.g., in search engines), and for specific downstream “readers” that benefit from access to the full passages (rather than simply the pre-condensed extractions).
>
>
> > **Some of the important implementation / training details can be moved to the main text from the appendix.**
>
> We thank you for this suggestion. If our work is accepted, we are eager to use the additional page to move more of the details about condensing (like the one you raised and that raised by reviewer EkjX), focused late interaction (raised by reviewer EV88), and LHO supervision (raised by reviewer EkjX) to the main text.

---

### Decision · Program_Chairs · 2021-09-27

**Decision:**

Accept (Spotlight)

**Comment:**

The paper propose a multi-hop reasoning system called Baleen based on the idea of iterative retrieval. It includes three main ideas: 1) condensed retrieval which summarizes the documents at each hop; focused late interaction which ranks the top-k scores and only includes those for later computation; latent hop ordering which learns to order the passages. The method is evaluated on two datasets, HotpotQA and  HoVer. Both experiments demonstrate the proposed method achieves significant better performance than baselines.

All reviewers agree on the quality of the work.  The performance is solid. The paper is clear and easy to follow. The authors may want to address the questions raised by reviewers. In addition, the reference part is rather nonstandard. Please cite the correct source, i.e. the official publication should be cited if it is published, instead of the arxiv version.